| Open Peer Review | Host-Microbial Interactions | Opinion/Hypothesis

# Microbial genotoxin-elicited host DNA mutations related to mitochondrial dysfunction, a momentous contributor for colorectal carcinogenesis

Xue Yang,[1,2] Yumeng Gan,[1,2] Yuting Zhang,[1] Zhongjian Liu,[3] Jiawei Geng,[1,2,4] Wenxue Wang[1,2,4]

**ABSTRACT** Gut microbe dysbiosis increases repetitive inflammatory responses, leading to an increase in the incidence of colorectal cancer. Recent studies have revealed that specific microbial species directly instigate mutations in the host nucleus DNA, thereby accelerating the progression of colorectal cancer. Given the well-established role of mitochondrial dysfunction in promoting colorectal cancer, it is reasonable to postulate that gut microbes may induce mitochondrial gene mutations, thereby inducing mitochondrial dysfunction. In this review, we focus on gut microbial genotoxins and their known and potential targets in mitochondrial genes. Consequently, we propose that targeted disruption of genotoxin transport pathways may effectively reduce the rate of mitochondrial gene mutations and yield substantial benefits for the prevention of colorectal carcinogenesis.

**KEYWORDS** colorectal cancer, gut microbes, genotoxins, DNA mutation

## Mitochondrial dysfunction in colorectal cancer

Mitochondria play an important role in various cellular physiological functions, including energy production (1), biosynthesis (2), oxidative processes (3), calcium signaling (4), programmed cell death (5), and metabolic signaling (6). Disturbances in mitochondrial function contribute to the development of cellular pathologies and promote the occurrence of diseases, such as colorectal cancer (CRC) (7–9). A typical and integral case is that succinate dehydrogenase (SDH)-coding gene mutation links mitochondrial dysfunction and carcinogenesis via interfering with hypoxia-inducible factor (HIF) protein stability (10–13). Consequently, the distinct features of mitochondria can be used to evaluate CRC risk and prognosis (8, 9, 14, 15).

## Mitochondrial DNA mutations and related mitochondrial dysfunction

Mitochondrial respiratory chains, responsible for energy production, subject mitochondrial DNA (mtDNA) to surrounding respiratory chain-produced reactive oxygen species (ROS). Meanwhile, mtDNA lacks both the protective histones and the complete DNA repair system (16, 17). These circumstances significantly enhance its susceptibility to mutations triggered by oxidative damage (18, 19). Such mutations can alter the copy number of mtDNA and perturb various processes, including oxidative phosphorylation (OXPHOS), tricarboxylic acid (TCA) cycle, and mitochondrial ROS production (3, 11). The accumulation of mtDNA mutations can lead to irreversible mitochondrial dysfunction and cellular malignant transformation (7, 9, 11, 20, 21).

Address correspondence to Jiawei Geng, Jiawei_Geng@kmust.edu.cn, jia_wei_geng@163.com, or Wenxue Wang, Wenxue.Wang@kmust.edu.cn, wenxue_wang@163.com.

The authors declare no conflict of interest.

See the funding table on p. 8.

## Microbial genotoxins directly elicit host nuclear DNA mutations and CRC carcinogenesis

Abundant colonization of the human gut by the microbiota exerts considerable effects on the intestinal barrier structure (22), nutritional metabolism (23, 24), and even physiological behavior of the host (25, 26), while microbial dysbiosis contributes to various gut diseases, such as inflammatory bowel disease (IBD) (27, 28) and CRC (29–31). Specific gut microbes not only stimulate intestinal cell proliferation, apoptosis, and inflammatory response but also directly induce nuclear DNA damage in the host via genotoxins, including cytolethal distending toxin (Cdt) (32, 33) and colibactin (34–36). Cdt, a heterotrimer of CdtA, CdtB, and CdtC, is typically secreted by pathogenic gram-negative bacteria, such as *Salmonella typhi* (33). Gut microbe-derived Cdt is transported into the host cell nucleus, where it induces single-strand (SSBs) or double-strand breaks (DSBs) in the genomic DNA, depending on its local concentration (34). Exposure to sublethal doses of Cdt impairs the DNA damage response, leading to impairment of genomic DNA and accumulation of DNA mutations, which collectively contribute to genomic sequencing errors and neoplastic transformation (37). Furthermore, colibactin, produced by enteropathogenic *Escherichia coli* (EPEC), also acts as a DNA-damaging agent that induces DSBs and genome instability, therefore, facilitating CRC initiation by regulating the cell cycle, apoptosis, and senescence (36, 38–41). Enterotoxigenic *Bacteroides fragilis* (ETBF) also produces protein toxins that induce chronic colitis and contribute to the development of CRC (42–44). Avirulence protein A from *Salmonella* strains (45, 46) and *Fusobacterium* adhesin A (FadA) from *Fusobacterium nucleatum* (47–49) can also promote CRC development by inducing host cell neoplastic transformation, see Fig. 1. However, the impact of microbe-produced toxins on the remaining host cellular DNA within mitochondria, which plays an essential role in cancer cell metabolism, is an area that requires further exploration.

## It is time to work on the possibility of microbial genotoxins to mutate mtDNA and facilitate colorectal carcinogenesis

Studies have shown that certain strains of *E. coli* (e.g., EPEC) reside in the human gut and initiate CRC through the actions of small-molecule genotoxins (39, 41). These genotoxins induce DNA mutations in the nuclear genome, leading to the neoplastic transformation of colorectal epithelial cells (38, 50). To date, however, no studies have explored whether genotoxins also give rise to mtDNA mutations and subsequent mitochondrial dysfunction, which may potentially facilitate CRC tumorigenesis, despite the presence of genotoxin-targeted sequences in mtDNA.

### There are genotoxin-targeted mtDNA sequences

Research has indicated that colibactin primarily targets the adenine base of the host nuclear DNA and induces DSBs via alkylation (51, 52). When the DNA repair system fails to adequately repair these DSBs, mutations occur within the host genome (36, 53). Furthermore, EPEC-secreted colibactin also demonstrates a high affinity for AT-rich hexameric motifs (38), which are commonly found in DSBs within the nuclear DNA of clinical CRC tissue (54). Considering these findings, we analyzed the entire human mtDNA sequence (16,569 bp) and identified 17 AT-rich hexameric motifs (AAAATT, AATTTT, and AAATTT) present in both the D-loop and protein-coding regions (Table 1). The protein-coding genes were distributed on electron transport chain complexes I, III, IV, and adenosine triphosphate (ATP) synthases (Table 1). In addition to colibactin, EPEC also secretes EspF and continuously consumes DNA mismatch repair (MMR) proteins MSH2 and MLH1, resulting in the dysfunction of the DNA MMR system and subsequent accumulation of irreparable mtDNA mutations (55). Taking into the above mtDNA sequence analysis, it is plausible that colibactin targets AT-rich hexameric motifs, resulting in DSBs within mtDNA and subsequent mitochondrial dysfunction upon colibactin entry into mitochondria (Fig. 1).

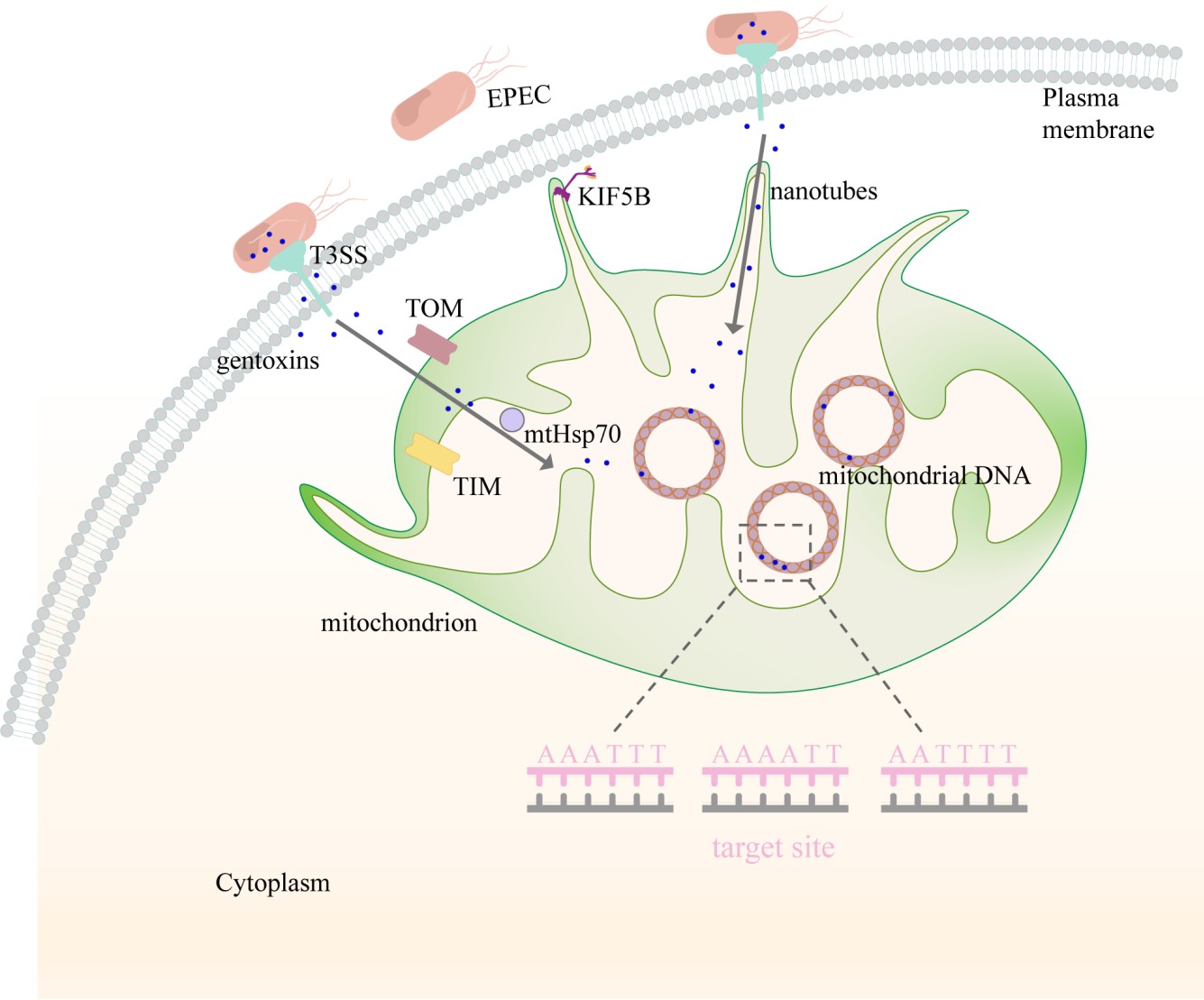

**FIG 1** Schematic model for microbial genotoxin translocation from extracellular matrix into mitochondria and consequent mtDNA damage (1). EPEC secrets genotoxins and injects them into host cellular cytoplasm via type III secretion system (T3SS). The injected genotoxins in cytoplasm can enter mitochondria under the assistances of (2) transport outer membrane (TOM), transport inner membrane (TIM), and matrix chaperone mtHsp70 proteins, or (3) KIF5B through pulling mitochondrial nanotubes to reach cellular membrane and connect T3SS (4). Then genotoxins inside mitochondria have an opportunity to directly bind to specific mtDNA sequences, such as AT-rich hexameric motifs (AAAATT, AATTTT, and AAATTT) (5). The genotoxins- induced mtDNA crack gives a contributions to cell malignant transformation, similar to nuclear DNA damage.

### Possible translocation pathway of genotoxins into mitochondria

Before the occurrence of microbe-induced mtDNA mutations, microbial effectors must first penetrate the cellular membrane to reach the cytoplasm. Following this, the effectors require additional assistance, such as the mitochondrial localization signal (MLS), to enable their translocation into the mitochondria rather than the nucleus (6).

### EPEC colibactin is transported into the host cellular cytoplasm through T3SS

Bacteria employ specialized secretion systems to facilitate the transport of their effectors into the cytosol of eukaryotic cells. Among these bacterial secretion systems, the most well-known system is the T3SS, serving as a transportation tunnel to enable the direct translocation of microbial effectors into eukaryotic cells, thereby disrupting the physiological processes of the host cell (56). In recent years, type VII secretion systems

**TABLE 1** AT hexamer distribution on mtDNA

| Genes | Region | Gene sequence range | AT hexamer | |
|---|---|---|---|---|
| | | | Sequence | Repeats |
| D-loop | D-loop | 289–294, 395–399 | AAATTT | 2 |
| | | 421–426 | TTTTAA | 1 |
| ND1 | | 3382–3387 | AAAATT | 1 |
| ND2 | | - | - | - |
| ND3 | | 10073–10078 | AATTTT | 1 |
| | | 10217–10222 | AAAATT | 1 |
| ND4 | Complex I | 10882–10887 | TTTTAA | 1 |
| | | 11465–11470 | TTAAAA | 1 |
| ND4L | | - | - | - |
| ND5 | | 3382–3387 | AAAATT | 1 |
| ND6 | | 14502–14507 | TTAAAA | 1 |
| | | 14536–14541 | AAAATT | 1 |
| Cytb | Complex III | 14779–14785 | AAAATT | 1 |
| COX1 | | 7307–7312 | AATTTT | 1 |
| COX2 | Complex IV | 8093–8098 | TTAAAA | 1 |
| COX3 | | 9704–9709 | AATTTT | 1 |
| ATP6 | ATP synthase | 8888–8893 | TTAAAA | 1 |
| ATP8 | | 8499–8504 | AAAATT | 1 |

have also attracted great interest for their functions and transport models (57–59). The translocation is influenced by various factors, such as bacterial effector density, effector-chaperone interactions, and bacterial wall-host cell membrane attachment (60). Three types of *E. coli*, that is, EPEC, Shiga toxin-producing *Escherichia coli* (STEC), and enteroinvasive *Escherichia coli* (EIEC), possess the T3SS (61), which they use to transport colibactin into host cells, effectively disrupting normal cellular processes and inducing a dysfunctional state that benefits their own survival and proliferation (60) (Fig. 1).

### Microbial effectors are modified to target host cell organelles

Bacterial effectors employ a diverse range of mechanisms to reach their final destinations and rely on various host cell-mediated processes, such as lipidation (62, 63), ubiquitylation (64, 65), and phospholipid binding (66–68), to anchor themselves to the host cytomembrane. Upon entry into the host cell cytoplasm, effectors undergo lipid attachment and covalent modifications facilitated by the host. This leads to an increase in the hydrophobicity of resident host proteins, which, in turn, promotes effector adherence, thus influencing their localization and function (69). Additionally, bacterial effectors can impact the ubiquitylation process, which serves as a signal for protein degradation, by acting as deubiquitinases or E3 ligases. This alteration in ubiquitylation ultimately affects the function and stability of host proteins (70–72). Notably, host protein ubiquitylation can determine the localization of bacterial effectors (64, 65). Furthermore, multiple bacterial effectors can target biochemically active sites of host proteins with the assistance of phosphoinositides (69).

### Microbial effectors target mitochondria via MLSs

The MLS plays a pivotal role in facilitating the localization of EPEC effectors within the mitochondria (Fig. 1). The EPEC effectors EspF, Map, and EspZ can enter the mitochondrial matrix with the assistance of the mitochondrial outer membrane translocases Tom22 and Tom40 and the matrix chaperone mtHsp70 (73, 74). Upon entry, the effectors induce alterations in mitochondrial membrane potential and morphology, thus influencing host cell functions (73, 75, 76). Specifically, Map guides bacterial effectors to the mitochondrial membrane via recognition of their N-terminal 44 residues by the TOM and TIM complexes. Once recognized, these complexes facilitate the crossing of

EPEC effectors through the mitochondrial outer membrane, thereafter reaching the mitochondrial matrix with the assistance of mtHsp70 (73, 77–79) (Fig. 1).

### Microbes could directly target mitochondria via "nanotubes"

Emerging evidence suggests that microbial nanotubes may function as a more convenient pathway for the translocation of effectors into host cells and organelles, such as mitochondria. Scanning electron microscopy (SEM) has revealed the presence of similar nanotubes in both bacteria (80–82) and mitochondria (6, 83, 84), which they use to connect neighboring cells and exchange cellular molecules. For instance, *E. coli* can rapidly establish connections with adjacent bacteria via protuberances on the cellular membrane, thereby accelerating the exchange of nutrients and genetic material (82). Similarly, mitochondria also form nanotubes to facilitate the exchange of genetic material (85–87). In our study, the coculturing of colorectal epithelial cells with EPEC revealed a significant enrichment of EPEC nucleic acid within the membrane protrusion structures of the epithelial cells, where a large accumulation of mitochondria was also observed. Concurrently, these colorectal epithelial cells generated many nanotube-like extensions, thus providing a material and spatial basis for EPEC to directly transport genotoxins into the mitochondria. Consequently, these bacterial nanotubes not only serve as conduits for gene and material exchange within microbial communities but also offer a direct channel for the transport of their genetic effects into mammalian cells and cellular mitochondria (Fig. 1).

### Microbial genotoxins may simultaneously insult nuclear DNA and mtDNA to favor CRC initiation

Numerous mtDNA mutations exist in CRC cells, leading to alterations in mitochondrial copy number, OXPHOS function, DNA repair systems, and cell cycle regulation (9, 11, 36). These mutations are tightly correlated with CRC progression and prognosis. For example, during the CRC initiation stage, mitochondrial pyruvate carrier (MPC) is found at extremely low levels, and its inactivation can induce tumorigenesis via the Wnt/β-catenin signaling pathway, thus serving as an independent risk factor for CRC (88). Mutations in mtDNA also lead to respiratory chain dysfunction and accumulation of intracellular ROS, thereby activating the oncogenic mitogen-activated protein (MAPK) and epidermal growth factor (EGF) signaling pathways (18). Moreover, mtDNA mutations can induce mitochondrial metabolic defects in fumarate, succinate, and 2-hydroxyglutarate, greatly promoting cell neoplastic transformation (89–91). Furthermore, low mitochondrial membrane permeability resulting from mtDNA mutations not only promotes cancer cell insensitivity to hostile environments but also increases tumor precursor cell tolerance to programmed cell death (92–94). Therefore, mtDNA mutations can induce various alterations in OXPHOS, energy metabolism, and ROS production, which intensively support cancer cell survival and metastasis (93, 95–97).

Microbes regulate the collaborations between mitochondrial genes and nuclear DNA in relation to arising carcinogenic performances. A typical case is succinate dehydrogenase deficiency, which leads to the succinate accumulation of in mitochondria and translocation into the cytosol. Succinate accumulation inhibits prolyl hydroxylases, which fail to degrade HIF-1α even under normoxic conditions. Consequently, stabilized HIF-1α binds to HIF-1β and forms HIF complex, which initiates transcription of nuclear genes involved in tumor progression (10, 98). Interestingly, Taylor and his colleagues discovered Porphyromonas somerae invades human cell under hypoxic conditions and suggested that this microbe may produce an excessive succinate, which is similar to the results of SDH gene mutation (12). Meanwhile, Jesper et. al., focused on Helicobacter pylori pathogenicity and revealed this microbe injects the virulence factors, cytotoxin-associated genes (CagA and CagE) and vacuolating cytotoxin (VacA) into the host epithelial cells. VacA encodes a mitochondrial targeting signal, guides virulence factors to translocate into mitochondria, and finally helps arise mitochondrial D-loop mutation (99). However, truncating the mutations of the mtDNA-encoded complex I

gene (MT-ND5) enhances checkpoint blockade response by driving aerobic glycolysis (100); even the this kind of mtDNA mutation induces mitochondrial dysfunction and colorectal carcinogenesis. The enhanced response may attribute to intensified cancerous imprint, such as aerobic glycolysis (Fig. 2).

Nevertheless, further studies are needed to establish whether microbial genotoxins induce mtDNA mutations and mitochondrial dysfunction and whether they collaborate with nuclear DNA mutation-induced abnormal cell characteristics, ultimately leading to the collective promotion of CRC initiation and progression.

## Concluding remarks

Despite the growing body of research focusing on CRC pathogenesis, several important questions remain to be answered, including whether microbes have the capacity to directly transport colibactin into the mitochondria and induce mtDNA mutations and CRC carcinogenesis (Fig. 3). A comprehensive understanding of the cellular transport system of colibactin would undoubtedly contribute to advancements in CRC prevention and therapy. There are several useful approaches to validate the proposed colibactin cellular transport system. First, after labeling colibactin with β-lactamase and enabling its fusion expression in EPEC, high-resolution confocal microscopy under co-culture

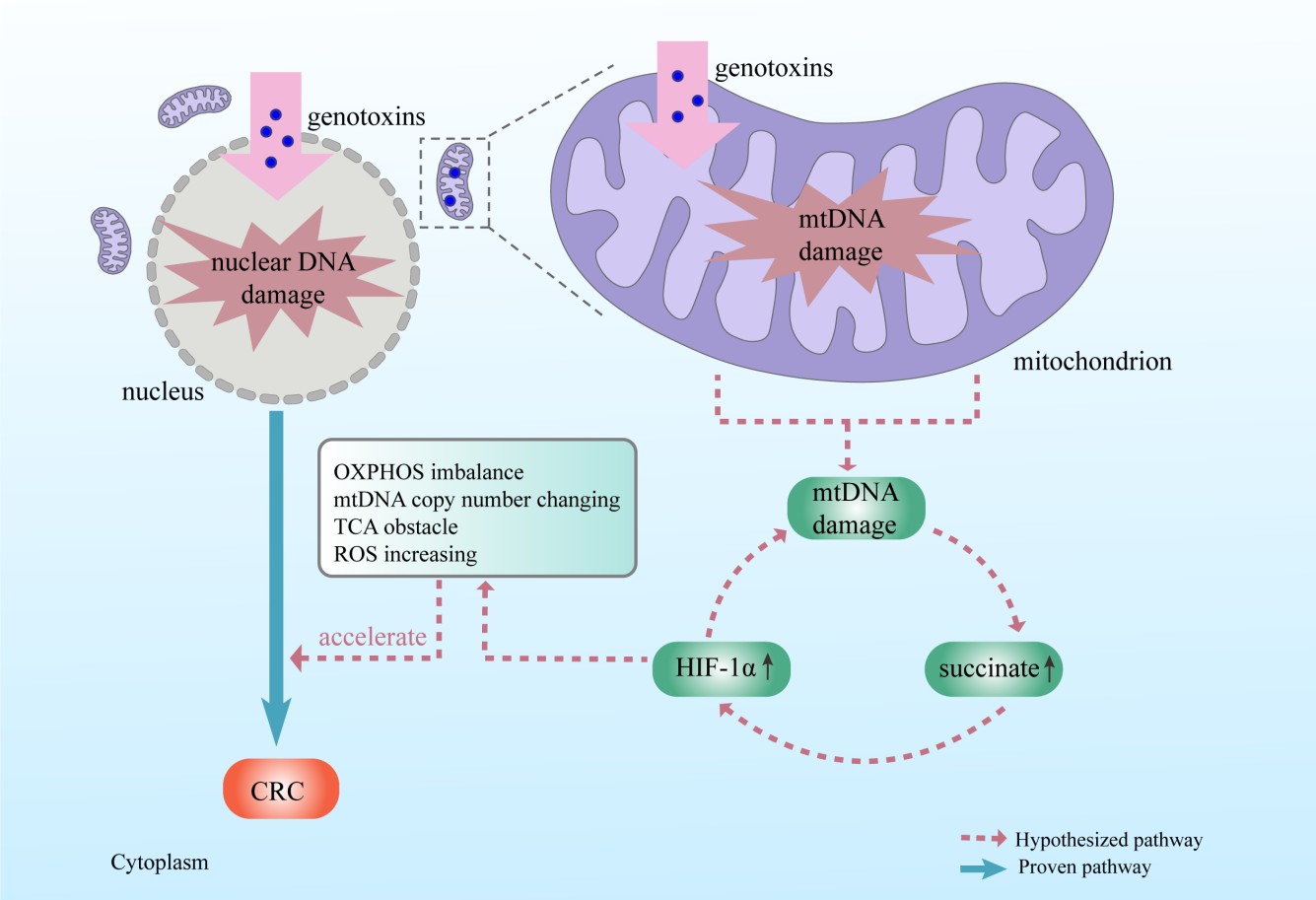

**FIG 2** Crosstalk between nuclear DNA damage and mtDNA damage, both elicited by microbial genotoxins and initiates colorectal carcinogenesis. Microbial genotoxins translocate into mitochondria and elicit mtDNA mutations, which induce succinate retention in mitochondria and following elevated HIF-1α expression. HIF-1α elevation further elicits mtDNA mutation via positive feedback. This wicked feedback causes OXPHOS imbalance, mtDNA copy number variation, TCA cycle obstacle, and excessive ROS production, all of which possess stimulative properties of colorectal carcinogenesis. HIF-1α, a transcription factor in response to hypoxia, plays a key role in balancing cellular oxygen levels and various biological processes. Succinate, an important intermediator of the TCA cycle and mitochondrial respiratory chain, is tightly correlated with mitochondrial function.

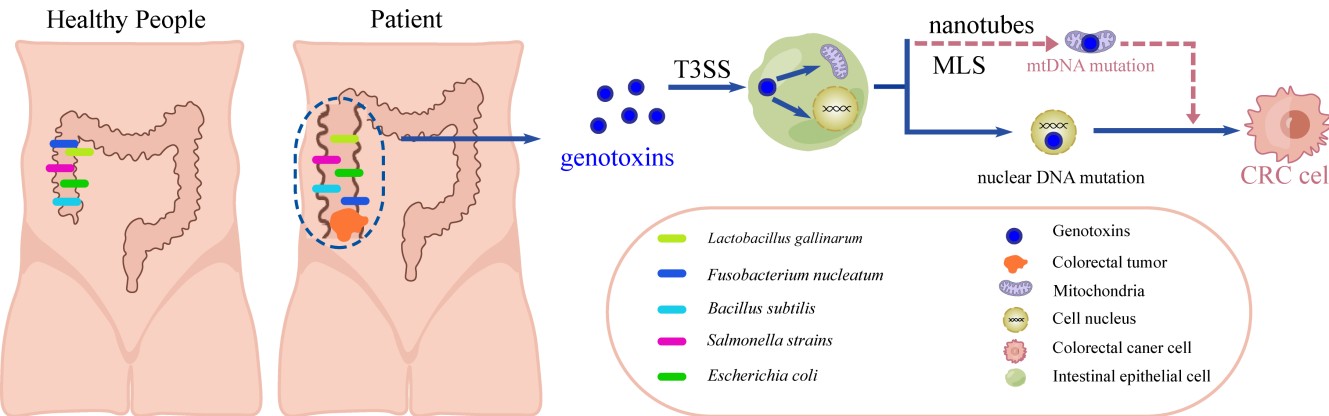

**FIG 3** Schematic model for microbial genotoxin-elicited CRC carcinogenesis. Gut microbes, such as *Lactobacillus gallinarum*, *F. nucleatum*, *Salmonella* strains, *Bacillus subtilis*, and EPEC, secret genotoxins and transport them into intestinal epithelial cytoplasm via T3SS and (or) "nanotube" under MLS guidance. Genotoxin induces the mutations of both nuclear DNA and mtDNA, and consequently accelerates colorectal carcinogenesis.

conditions of EPEC and mitochondria could be used to observe whether colibactin is indeed transported into the mitochondria. Nanotube disruption and reconstruction, along with MLS gene knockout, represent potential strategies that could offer compelling evidence to validate the aforementioned proposal. SEM and transmission electron microscopy (TEM) are useful tools to capture these disruption and reconstruction processes. Of note, there are currently no reports on fungus- or virus-produced genotoxins and their correlations with CRC, which could provide novel insights into CRC carcinogenesis.

## Glossary

**ATP:** a universal mediator of metabolism and signaling across unicellular and multicellular species.

**Cancer stem-like cells (CSCs):** the cancer cells with the properties of stem cells, having the ability to self-renewal and multicellular differentiation.

**CRC:** a common malignant tumor in the gastrointestinal tract, usually accompanied by bloody stool, diarrhea, and abdominal pain.

**Cdt:** an AB-type bacterial toxin, causing DNA DSB reaction and consequent irreversible cell cycle arrest and cell apoptosis.

**DSBs:** dual strands of DNA chain break simultaneously at the same corresponding or adjacent location.

**EPEC:** an important cause of childhood diarrhea, characterized by the presence of the intimin adhesin and the bundle-forming pilus.

**ETBF:** usually causes diarrhea and colitis and increases cancer risk after their durative insult to intestinal mucosa.

**EIEC:** also causes bacillary dysentery, is better adapted to survive in the environment and not to be restricted to host dependency for survival.

**EGF:** a conventional mitogenic factor that stimulates the proliferation of various types of cells including epithelial cells and fibroblasts.

**FadA:** a pili adhesion protein that is essential for bacterial attachment and invasion of gingival epithelial and endothelial cells.

**HIF:** transcriptional factors that respond to hypoxia, including HIF-1α and HIF-2α, play a key role in maintaining cellular oxygen balance and regulating multiple biological processes.

**IBD:** a group of inflammatory conditions of the colon and small intestine, Crohn's disease, and ulcerative colitis being the principal types.

**MMR:** has a major role in the detection and correction of DNA replication errors, resulting from DNA polymerase slippage or nucleotides misincorporation.

**mtDNA:** a maternally inherited DNA located in the mitochondria of eukaryotic cells.

**MLS:** a short peptide, about 15–70 amino acids long, bearing positively charged basic residues, that directs the transport of a protein to the mitochondria.

**MPC:** an inner mitochondrial membrane complex that plays a critical role in intermediary metabolism.

**MAPK:** a group of evolutionarily highly conserved ubiquitous proteins that have significant function as facilitators of signal transduction.

**OXPHOS:** a biochemical process occurs in the mitochondrial membrane of eukaryotic cells or the cytoplasm of prokaryotes, along with the ATP generation.

**ROS:** derived from both nicotinamide adenine dinucleotide phosphate oxidase (NOX) and mitochondria play a critical role in many physiological and pathological processes.

**SDH:** a tumor suppressor gene, responsible for converting succinate to fumarate, which releases electrons as part of the citric acid cycle and provides an attachment site for released electrons to be transferred to the OXPHOS pathway.

**SSBs:** one strand of DNA breaks in the DNA double helix structure.

**TIM:** the home for the proteins of OXPHOS, and within the membrane, the proteins are very closely packed.

**TOM:** a double phospholipid membrane that separates the intermembrane space from the cytosol.

**TCA:** the final common oxidative pathway for carbohydrates, fats, and amino acids.

**Tumor microenvironment (TME):** the surrounding microenvironment, in which tumor cells exist, including surrounding blood vessels, immune cells, fibroblasts, bone marrow-derived inflammatory cells, various signaling molecules, and extracellular matrix.

## ACKNOWLEDGMENTS

This study was supported by a grant from the National Natural Science Foundation of China (82160514, 82360395) and the Foundation of First People's Hospital of Yunnan Province (KHYJ-6-2020-001, KHBS-2022-028, 2022-KHRCBZ-C06).

## AUTHOR AFFILIATIONS

[1]Department of Infectious Disease and Hepatic Disease, The Affiliated Hospital of Kunming University of Science and Technology, The First People's Hospital of Yunnan Province, Kunming, Yunnan, China

[2]School of Medicine, Kunming University of Science and Technology, Kunming, Yunnan, China

[3]Institute of Basic and Clinical Medicine, First People's Hospital of Yunnan Province, Affiliated Hospital of Kunming University of Science and Technology, Kunming, Yunnan, China

[4]Faculty of Life Science and Technology, Kunming University of Science and Technology, Kunming, Yunnan, China

## AUTHOR ORCIDs

Jiawei Geng http://orcid.org/0000-0002-9122-1532
Wenxue Wang http://orcid.org/0000-0002-5057-1851

## FUNDING

| Funder | Grant(s) | Author(s) |
| --- | --- | --- |
| MOST | National Natural Science Foundation of China (NSFC) | 82160514 | Wenxue Wang |
| MOST | National Natural Science Foundation of China (NSFC) | 82360395 | Wenxue Wang |

## ADDITIONAL FILES

The following material is available online.

### Open Peer Review

**PEER REVIEW HISTORY (review-history.pdf).** An accounting of the reviewer comments and feedback.

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
