## [Reviewer comments · mSystems]

Microbial genotoxin-elicited host DNA mutations related to mitochondrial dysfunction, a momentous contributor for colorectal carcinogenesis

Xue Yang, Yumeng Gan, Yuting Zhang, Zhongjian Liu, Jiawei Geng, and Wenxue Wang

Corresponding Author(s): Jiawei Geng, First People's Hospital of Yunnan Province

Review Timeline:

Submission Date:

July 30, 2024

Accepted:

July 31, 2024

Editor: Nicholas Chia

Reviewer(s): The reviewers have opted to remain anonymous.

Transaction Report:

DOI: <https://doi.org/10.1128/msystems.00887-24>

Re: mSystems00887-24 (Microbial genotoxin-elicited host DNA mutations related to mitochondrial dysfunction, a momentous contributor for colorectal carcinogenesis)

Dear Prof. Jiawei Geng:

I find this manuscript to be much improved and am therefore making an editorial decision to accept.

Your manuscript has been accepted, and I am forwarding it to the ASM production staff for publication. Your paper will first be checked to make sure all elements meet the technical requirements. ASM staff will contact you if anything needs to be revised before copyediting and production can begin. Otherwise, you will be notified when your proofs are ready to be viewed.

Sincerely,
Nicholas Chia
Editor
mSystems